# Do Boat and Ocean Suggest Beach?
# Dialogue Summarization with External Knowledge

**Tianqing Fang**                                                        TFANGAA@CSE.UST.HK
**Haojie Pan**                                                       HPANAD@CONNECT.UST.HK
**Hongming Zhang**                                                      HZHANGAL@CSE.UST.HK
**Yangqiu Song**                                                         YQSONG@CSE.UST.HK
*Department of Computer Science and Engineering, HKUST, Hong Kong, China*

**Kun Xu**                                                            KXKUNXU@TENCENT.COM
**Dong Yu**                                                               DYU@TENCENT.COM
*Tencent AI Lab, Bellevue WA, USA*

## Abstract

In human dialogues, utterances do not necessarily carry all the details. As pragmatics studies suggest [Grice, 1975], humans can infer meaning from the situational context even though the meaning is not literally expressed. It is crucial for natural language processing models to understand such an inference process. In this paper, we address the problem of inferring Concepts Out of the Dialogue Context (CODC) in the dialogue summarization task. We propose a novel framework comprised of a CODC inference module leveraging external knowledge from WordNet and a knowledge attention module aggregating the inferred knowledge into a neural summarization model. To evaluate the inference capability of different methods, we also propose a new evaluation metric based on CODC. Experiments suggest that current automatic evaluation metrics of natural language generation may not be enough to understand the quality of out-of-context inference in generation results, and our proposed summarization model can provide statistically significant improvements on both CODC inference and traditional automatic evaluation metrics, e.g., CIDEr. Human evaluation of the model's inference ability also demonstrates the superiority of the proposed model. Codes and data are available at https://github.com/HKUST-KnowComp/CODC-Dialogue-Summarization.

## 1. Introduction

Automatically summarizing conversations in our daily lives can benefit users for better organizing and retrieving their historical information. There have been several approaches to conversation summarization, including extractive approaches [Xie et al., 2008, Riedhammer et al., 2010] and abstractive approaches [Oya et al., 2014, Shang et al., 2018]. While extractive approaches focus on using the seen words in a conversation to summarize it, abstractive approaches usually use a text generation model to perform summarization.

Different from news articles, in daily dialogues, it is common that a speaker's utterance can suggest something that is not literally expressed but can be interpreted by a cooperative listener. For example, in Figure 1, if a dialogue mentions *boat* and *ocean*, the first impression of an ordinary person would be a boat sailing on the ocean. If, a modifier *abandoned* is added to *boat*, combined with *ocean*, the previous scene will be canceled and we will think of a beach or a shore because that is where the abandoned boat and ocean tend to cooccur. This kind of out-of-context inference is a language phenomenon in the field of pragmatics

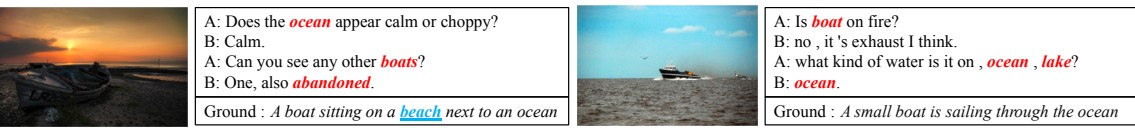

Figure 1: Two examples in DialSum. We highlight contextual concepts and out-of-context inference with **bold italics** and **underlined bold italics** respectively. "Ground" indicates ground truth summarizations.

[Grice, 1975]. To comprehensively understand a dialogue where some context is omitted by the speakers, both a thorough understanding of the context and necessary inference are required. However, to the best of our knowledge, neither evaluation metrics nor suitable methods are available for such kind of out-of-context inference currently, even if it is an easy task for human beings.

To study the phenomenon of out-of-context inference in dialogues, we narrow down the definition of out-of-context inference to a certain type so that we can automatically evaluate it. Here, we only focus on summarization's *new concepts* that are suggested by existing concepts in the dialogue context. This is also related to *lexical entrainment* [Brennan, 1996] which studies the lexical variability in language use and *lexical pragmatics* in relevance theory [Sperber and Wilson, 1986], which studies how to identify and infer concepts from words via broadening and narrowing contexts. To formally define the inference of new concepts in summarization, we distinguish a new concept from existing concepts following three rules according to WordNet [Miller, 1995]: (1) It should not be a general concept. (2) It should not be a synonym of an existing concept. (3) It should not be a hypernym (or super-concepts) of an existing concept. In this way, we can distinguish out-of-context inference from logical entailment or synonym as much as we can. Specifically, we name them as the Concept Out of the Dialogue Context (CODC.) Current evaluation metrics are usually based on lexical similarities or overlaps, e.g., BLEU [Papineni et al., 2002], ROUGE-L [Lin, 2004], METEOR [Lavie and Agarwal, 2007], CIDEr [Vedantam et al., 2015], etc. Such lexical metrics usually do not care about the recall of the novel words, and are not precise at evaluating the precision of the out-of-context inference. To tackle these limitations, we propose and study a new evaluation metric that incorporates CODC for text generation.

Moreover, to improve the summarization results with the help of CODC inference, we propose an abstractive summarization framework that includes an out-of-context inference module to enhance the model's inference ability. The two-step framework we proposed basically contains an inference module and an extendable knowledge attention model. In the inference part, we use word-relatedness features such as co-occurrence, embedding similarities, and WordNet relatedness features to distinguish whether a candidate word is a plausible out-of-context inference or not. While in the knowledge attention module, we aggregate the retrieved knowledge from the inference module and apply the attention mechanism to extract useful information in the decoding steps.

Our contributions are summarized as follows:

1. We address the problem of out-of-context inference in dialogue summarization for the first time, and provide an elaborative definition of the problem.

2. We design a related metric based on Concepts Out-of Dialogue Context (CODC) to evaluate neural models' ability to infer plausible novel concepts.

3. We proposed Trans-KnowAttn, an abstractive dialogue summarization approach that incorporates an out-of-context inference (missing-link inference) module and a knowledge attention module to improve the inference capability in dialogue summarization.

## 2. Dialogue Summarization Task

We develop our dialogue summarization (DialSum) task based on the VisDial [Das et al., 2017] dataset. To construct their dataset, VisDial asks two workers on Amazon Mechanical Turk to chat with each other in real-time to discuss an image in the MSCOCO dataset [Lin et al., 2014]. The MSCOCO dataset contains human-annotated captions of about 120K images. Each image has five captions from five different annotators. A "questioner" sees the caption of the image and another person sees both the caption and the image. The questioner is asked to ask questions to "imagine the scene better", and the annotators usually describe the most prominent concepts in an image. Thus, for each image, we have both a dialogue from the VisDial dataset and five captions from the MSCOCO dataset. We align a dialogue with five captions as five alternative summarizations of the dialogue. By nature, there are many out-of-context inference phenomena behind those utterances, as the speakers already have the image in mind as their context so that they don't need to bring it up again in the dialogue. Assuming a gold summarization $y$ is provided by an annotator, we evaluate models' inference ability by how many novel concepts (i.e., noun phrases that *do not* appear in the dialogue) in $y$ can be mentioned by $\hat{y}$, the generated summary, without introducing extra noisy concepts. The number of examples in the training, developing, and testing set are 98,256, 12,282, and 12,083, respectively.

## 3. CODC-based Evaluation Metric

### 3.1 Definition of CODC

In this section, we introduce the new metric based on CODC for evaluating models' conceptual inference ability. Here, by concept, we mean a noun word. As most examples in our dataset are general objects rather than specific named entities, we use WordNet [Miller, 1995] to check the *relations* among all the concepts.

For each example in DialSum, which contains a dialog $x$ and a description $y$, we denote the concept sets extracted from $x$ and $y$ as $\mathcal{C}^x$ and $\mathcal{C}^y$ respectively. Then we define the set of concepts out of dialogue context as

$$\mathcal{C}^{y-x} = \{c_y \in \mathcal{C}^y | \forall c_x \in \mathcal{C}^x, I_1(c_x, c_y) = 1)\}, \tag{1}$$

where $I_1(c_x, c_y)$ is the function implementing the following three rules:
• The least depth of all synsets containing $c_y$ is no less than $\delta_d$, where we set $\delta_d = 4$ empirically (1,790 synsets in total are filtered out.) This rule is to filter out very general concepts that may be inferred by anything, for example, "entity".
• The shortest path between any pair of synsets containing $c_x$ and $c_y$ is greater than 1. This rule is to filter out concepts that are synonyms of existing concepts in the dialogue.

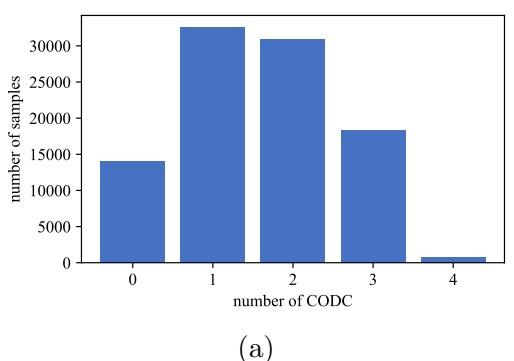
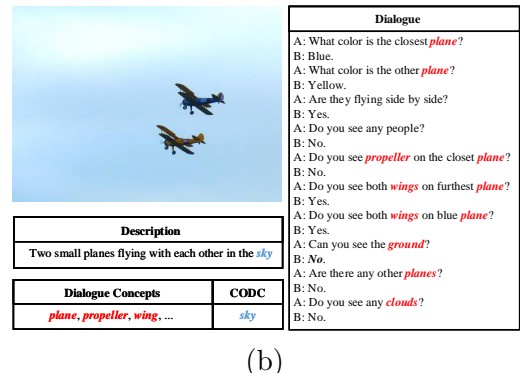

(a)                                                    (b)

Figure 2: (a) Distribution of CODC. (b) An example of out-of-context inference in the dataset. We also show the corresponding figures from MSCOCO for better demonstration. The figures are not used in our experiments.

• Any synset of $c_y$ is not a hypernym (super-concept) of any synset of $c_x$. This rule is to avoid entailment for the new concepts as essentially abstractive summarization is able to perform conceptual abstraction.

To better understand the dataset, we show the distribution of the counts of CODC per dialogue in Figure 2 (a), for one set of the descriptions in the training set. We can find that for more than 80% of the training dialogues, one or more CODC should be inferred. In Figure 2 (b), we show a real example from the dataset, where we can infer the word *sky* from the dialogue even if the word is not literally expressed. These observations show that the out-of-context concepts are common in this dialogue summarization task and further prove the importance of understanding how well models can generate summarization with out-of-context concepts.

### 3.2 CODC Precision, Recall, and F1

As a model with strong inference ability should be able to infer correct new concepts without introducing wrong ones, inspired by the F1 evaluation metric, we design F1 over CODC to evaluate models' inference ability. Let $\mathcal{X} = \{x^{(1)}, x^{(2)}, ..., x^{(N)}\}$ be all dialogues in the evaluation set, where $N$ is the size of the dataset, and $\mathcal{Y} = \{\mathcal{Y}^{(1)}, \mathcal{Y}^{(2)}, ..., \mathcal{Y}^{(N)}\}$ be the set of ground-truth descriptions. Note that, for each image, we may have $K$ ground-truth descriptions such that $\mathcal{Y}^{(i)} = \{y_1^{(i)}, ..., y_K^{(i)}\}$. Denote the generated summary of the $i$-th dialogue as $\hat{y}^{(i)}$. We then define the precision, recall, and F1 over CODC as follows:

$$P_{CODC} = \frac{\sum_{i=1}^{N} \max_k |h(x^{(i)}, y_k^{(i)}, \hat{y}^{(i)})|}{\sum_{i=1}^{N} |\mathcal{C}^{\hat{y}^{(i)} - x^{(i)}}|},$$

$$R_{CODC} = \frac{\sum_{i=1}^{N} |h(x^{(i)}, y_{k_i^*}^{(i)}, \hat{y}^{(i)})|}{\sum_{i=1}^{N} |\mathcal{C}^{y_{k_i^*}^{(i)} - x^{(i)}}|}, \tag{2}$$

$$F1_{CODC} = \frac{2 P_{CODC} R_{CODC}}{P_{CODC} + R_{CODC}},$$

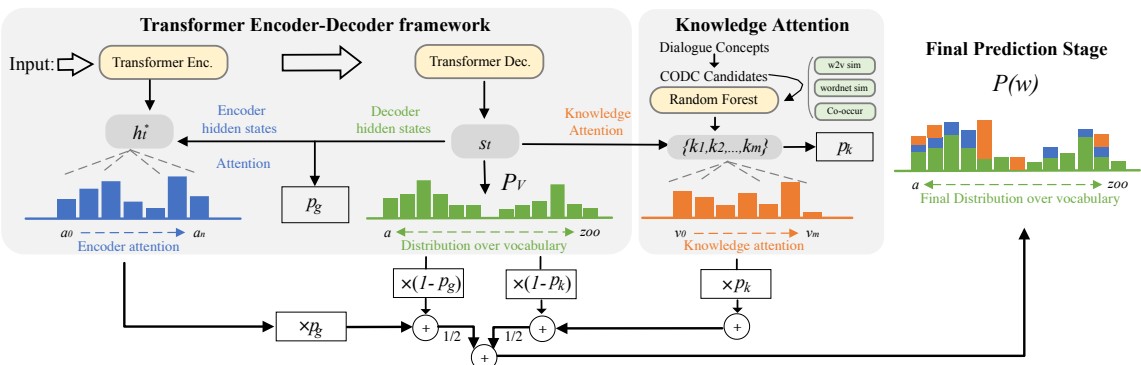

Figure 3: Overview of the Trans-KnowAttn framework.

where $h(x, y, \hat{y})$ and $k_i^*$ are defined as follows:

$$h(x, y, \hat{y}) = \{c_{\hat{y}} \in \mathcal{C}^{\hat{y}-x} | \exists c_y \in \mathcal{C}^{y-x}, I_2(c_y, c_{\hat{y}}) = 1\}, \tag{3}$$

$$k_i^* = \operatorname{argmax}_{k \in \{1, \dots, K\}} \frac{|h(x^{(i)}, y_k^{(i)}, \hat{y}^{(i)})|}{|\mathcal{C}^{y_k^{(i)} - x^{(i)}}|}, \tag{4}$$

and $I_2(c_y, c_{\hat{y}})$ is the function to determine whether $c_y$ and $c_{\hat{y}}$ are identical or entailed following either of two rules:

• There exists one pair of synsets of $c_y$ and $c_{\hat{y}}$ being identical.

• There exists a synset of $c_{\hat{y}}$ being a hypernym of a synset of $c_y$.

Note that a summary can be considered as a plausible generation if it covers a reasonable amount of CODCs for **one of** the $K$ descriptions. Based on this idea, for a certain dialogue, we only focus on the description where the generated sentence performs the best in precision or recall regarding the inference of CODC. In this case, *max* operation is selected in the calculation of precision and recall, which is different from the *average* operation that is typically used by previous automatic evaluation metrics in the setting of multi-reference. Also, by using the *max* operations, we argue that the theoretical upper bound for CODC precision, recall, and F1 are 1.0, as any ground-truth description will be scored 1.0 under the *max* operation.

## 4. Knowledge-aware Summarization

In this section, we present Trans-KnowAttn, a knowledge-aware summarization framework which consists of a missing-link inference module and a flexible knowledge attention module that can be applied to general encoder-decoder models. An overview of the model is shown in Figure 3.

### 4.1 Missing-link Inference Module

The missing-link inference module infers plausible concepts closely related to the concepts mentioned in the dialogue, while being out of the dialogue context. For example, considering the dialogue and ground-truth description in Figure 1, given *beach* and *boat* that are

mentioned in the conversation, we find a list that potentially contains the concept *beach*. This process is different from simple Knowledge Base Completion in that we want to infer the missing links between a concept with a set of dialogue concepts.

Following the definitions in the CODC metric, we first build a bipartite knowledge graph $G = (V, E), V = (D, C)$, that records the co-occurrence information of dialogue concepts and CODCs in the training set. Here $D$ represents the set of concepts in all training dialogues, and $C$ represents the set of concepts in CODCs. The weight of each vertex $u(c_x), c_x \in D$ or $u(c_y), c_y \in C$ is assigned by their total number of occurrences in the training set. An edge $(c_x, c_y), c_x \in D, c_y \in C$ exists if there are at least one dialogue-description pairs such that $I_1(c_x, c_y) = 1$, as has been defined in Equation (1.) The weight of the corresponding edge $u(c_x, c_y)$ is the number of co-occurrence of the concept pair in the training set. Based on the formulation of the co-occurrence graph, we formalize the inference process as follows:

1. Extract all concepts using rules defined in Section 3 from a dialogue as $c_x^{(1)}, c_x^{(2)}, \ldots, c_x^{(k)}$, where $k$ is the number of extracted concepts for dialogue $x$.

2. Get the set $\cup_{i=1,\ldots,k} N(c_x^{(i)})$ as primitive knowledge candidates, where $N(c_x^{(i)})$ is the set of neighbors of vertex $c_x^{(i)}$ in $G$.

3. Calculate features of all candidates and train a classifier to determine whether a candidate is a plausible CODC or not. The ground-truth CODCs are calculated based on the definition in Section 3.

4. In the inference process, use the classifier from the above step and retrieve top $m$ results ranked by the classifier, among the set $\cup_{i=1,\ldots,k} N(c_x^{(i)})$.

The features we use are described as follow:

**Co-occurrence.** A simulated co-occurrence feature of a candidate $n \in \cup_{i=1,\ldots,k} N(c_x^{(i)})$ given a set of dialogue concepts $\{c_x^{(1)}, \ldots, c_x^{(k)}\}$ is defined as : $P_{co}(n|\{c_x^{(1)}, \ldots, c_x^{(k)}\}) \propto \sum_{i=1}^{k} \log[\frac{u(n, c_x^{(i)})}{u(c_x^{(i)})}]$ , which is an ad-hoc attribute that depicts the log probability that a candidate concept $n$ will co-occur given all dialogue concepts.

**Pre-trained Word Embedding Similarities.** We compute the average word embedding similarities of candidate concept $n$ with all the dialogue concepts $\{c_x^{(1)}, \ldots, c_x^{(k)}\}$ as : $\frac{1}{k} \sum_{i=1}^{k} \frac{\mathbf{e}_{c_x^{(i)}} \cdot \mathbf{e}_n}{||\mathbf{e}_{c_x^{(i)}}|| \cdot ||\mathbf{e}_n||}$ , where $\mathbf{e}_{c_x^{(i)}}$ and $\mathbf{e}_n$ are embedding vectors of $c_x^{(i)}$ and $n$ obtained from existing pretrained vectors such as Word2Vec [Mikolov et al., 2013] and Glove [Pennington et al., 2014].

**WordNet Synset Relatedness.** We choose some typical similarity measurements based on WordNet as an indicator of relatedness, Path Similarity, i.e., the inverse of the number of nodes visited in the path from one word to another via hypernym hierarchy, LCH Similarity [Leacock et al., 2002], and WuP Similarity [Wu and Palmer, 1994].

### 4.2 Knowledge Attention Network

Treating the inferred candidate list as knowledge, we provide a knowledge attention mechanism that can be added to general encoder-decoder frameworks. We choose Transformer as the base architecture of the seq2seq model. To generate summaries, the decoder computes

| Method | BLEU-4 | METEOR | ROUGE-l | CIDEr | $P_{CODC}$ | $R_{CODC}$ | $F1_{CODC}$ |
|---|---|---|---|---|---|---|---|
| BertSum | 23.49 | 22.89 | 49.38 | 79.94 | 36.48 | 38.90 | 37.65 |
| S2S-Attn | 29.90 | 24.51 | 52.45 | 96.55 | 44.32 | 42.46 | 43.37 |
| PGN | 30.12 | 24.58 | 52.66 | 97.97 | 45.36 | 42.49 | 43.88 |
| Pair-encoder | **31.26** | 25.34 | 53.26 | 101.04 | 45.10 | 44.39 | 44.74 |
| Trans-Copy | 31.09 | 25.54 | 53.38 | 102.81 | 46.20 | 44.55 | 45.36 |
| Trans-KnowAttn | 31.22 | **25.93** | **53.70** | **104.00**$^*$ | **46.31** | **45.66**$^*$ | **45.98**$^*$ |

Table 1: Evaluations on conventional metrics and CODC metrics are presented, where bold scores are the best among all models. $^*$ after bold figures indicates the improvements are statistically significant with $p < 0.05$.

the hidden states and attends to the knowledge embedding list step by step, adjusting the current decoder state with the help of the attended knowledge vector. Also, a copy mechanism is used for copying useful candidate words directly from the knowledge candidate list. The model differs from the standard attention encoder-decoder framework in that an extra layer of knowledge attention is added to the decoding part. More detailed formulations of the summarization model are provided in Appendix A.

## 5. Experiments

### 5.1 Baselines

We select S2S-Attn [See et al., 2017], PGN [See et al., 2017], Trans-Copy [Vaswani et al., 2017], PairEncoder [Pan et al., 2018], and BertSum [Liu and Lapata, 2019] as baseline models:

**S2S-Attn** is a typical sequence-to-sequence model with attention mechanism [See et al., 2017], where the encoder is a single-layer bi-LSTM and the decoder is a single-layer unidirectional LSTM.

**PGN** is a hybrid pointer-generator network that can copy words from the source text via pointing mechanism [See et al., 2017].

**Trans-Copy** uses the Transformer network [Vaswani et al., 2017], which incorporates the self-attention mechanism in both encoder and decoder, and uses the copy mechanism for the decoder.

**PairEncoder** [Pan et al., 2018] is a model particularly developed for the same problem as ours, where a modified encoder based on the Transformer network is developed to emphasize the interaction between two speakers.

**BertSum** [Liu and Lapata, 2019] uses the pre-trained BERT [Devlin et al., 2019] as the encoder, and Transformer as the decoder.

### 5.2 Experimental Settings

**Inference Module:** In the inference module, a Random Forest classifier from sklearn [1] is used to distinguish whether a candidate word is a plausible CODC or not, and we select top $m = 13$ candidates ranked by the classifier as the knowledge that is fed into the knowledge

---

1. https://scikit-learn.org

| | General Quality | OOC-Inference |
|---|---|---|
| Trans-Copy | 0.484 | 0.473 |
| Trans-KnowAttn | **0.516** | **0.527** |

| | $P_{CODC}$ | $R_{CODC}$ | $F1_{CODC}$ |
|---|---|---|---|
| all | 8.83 | 46.71 | 14.85 |
| -glove | 9.21 | 48.48 | 15.48 |
| -w2v | **9.30** | **48.91** | **15.63** |
| -cooccurrence | 5.84 | 30.13 | 9.78 |
| -wordnet | 8.62 | 45.66 | 14.51 |

Table 2: (a) Results of human evaluation. OOC-inference indicates out-of-context inference. (b) Effects of different features. - before the name of the features indicates removing this feature in the classification.

attention module. Details of the ablation study of this classification task are shown in Section 5.5.

**Knowledge Attention Summarization Model:** For all models except for PairEncoder, the input sequences are concatenated dialogue text with marks of `<q>` and `<a>` to identity different turns with two speakers. For PairEncoder, the input is a list of utterance pairs. All five references are used in the training process. Each dialogue is used in five training instances accompanied with the corresponding five captions. For models except for BertSum, we use a vocabulary of 20K words out of in total 28K words. For all RNN-based models, 256-dimensional RNN hidden states and 256-dimensional word embeddings are applied for both knowledge word embeddings and encoder-decoder embeddings. For Trans-Copy, the dimension of word embeddings is set to be 256. For the BertSum model, we follow the same experimental setting as its original paper [Liu and Lapata, 2019].

### 5.3 Evaluation

**Automatic Evaluation**: Besides CODC metrics, conventional lexical metrics BLEU [Papineni et al., 2002], ROUGE-L [Lin, 2004], METEOR [Lavie and Agarwal, 2007], and CIDEr [Vedantam et al., 2015] are used for evaluation.

**Human Evaluation**: To better understand the effects of the knowledge attention module, we conducted human annotation on the overall quality and the out-of-context inference (missing-link inference) ability for 100 randomly selected summaries, generated by Trans-Copy and Trans-KnowAttn, respectively. Each pair of summaries were evaluated by 5 annotators. We run the evaluation using workers from Amazon Mechanical Turk (AMT.) Workers were given full illustrations about the definition of out-of-context inference and were asked to judge which summarization is better in terms of overall abstraction quality and out-of-context inference ability. The summaries were randomly shuffled to prevent unexpected bias. There was also a neutral option for the annotators if they find no difference between the two given summaries.

### 5.4 Results and Evaluation

Results of conventional metrics and CODC metrics are shown in Table 1. For the main metric, CIDEr, there's a 1.2 points improvement from Trans-Copy to Trans-KnowAttn. In addition, the improvements on the CODC Recall and F1 are also significant, indicating that the inferential ability of the model is improved by the inference module. We carry out

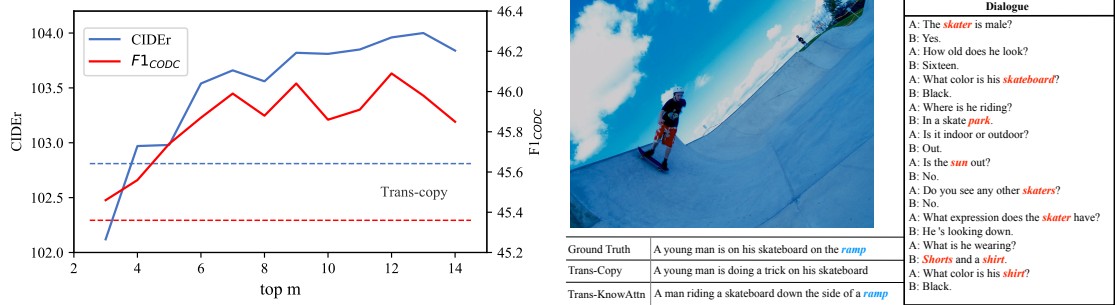

Figure 4: (a) Effects on the performance of the Trans-KnowAttn model given different number of retrieved knowledge words $k$. Dashed lines indicates the corresponding scores of Trans-copy baseline. (b) A case of CODC inference. We highlight contextual concepts and CODC inference with **bold italics** and **bold italics** respectively.

a statistical test using Randomization Test [Cohen, 1995]. All the improvements except for BLEU-4 and $P_{CODC}$ are statistically significant. The human evaluation results are presented in Table 2 (a), where the scores for each model are proportional to the number of times a model is considered superior to the other. The scores in each columns are normalized so that the sum equals 1. We show that Trans-KnowAttn brings consistent improvements in terms of both inference ability and overall summarization quality.

## 5.5 Ablations

**Inference Module:** We check the importance of different features by removing each of them in the random forest classifier. As shown in Table 2 (b), we report the retrieval results on the validation set under top 10 retrieval, and the CODC scores are calculated on the retrieved knowledge. It shows that keeping one of the w2v or glove similarities would boost the retrieval performance. Also, the co-occurrence feature is the most prominent one among all. In the end, we select the -w2v setting for the summarization.

**Summarization Results:** We study the overall $F1_{CODC}$ and CIDEr scores of the generated summaries, given the number of retrieved knowledge words that are fed into the knowledge attention model, as shown in Figure 4 (a). When the number of retrieved words $m$ exceeds 4, the overall CIDEr score can outperform the Trans-Copy baseline. For $F1_{CODC}$, it reaches the maximum when $m = 12$.

## 5.6 Case Study

We show an example of the generated results in Figure 4 (b). We observed concepts such as "*skater*," "*skateboard*," and "*skate park*" from the dialogue, from which a human will easily infer a "*ramp*" from the context which is a common environment where people play skateboards. The Trans-Copy model fails to infer such a new but highly relevant concept while Trans-KnowAttn will successfully yield this inference. More case studies will be presented in the appendix.

## 6. Related works

We introduce the related work in two-fold: neural text summarization and pragmatics tasks in NLP.

### 6.1 Neural Text Summarization

Conversation summarization has been studied extensively. It can be extractive approaches [Zechner, 2001, Nenkova and Bagga, 2003, Maskey and Hirschberg, 2005, Xie et al., 2008, Riedhammer et al., 2010] and abstractive approaches [Oya et al., 2014, Banerjee et al., 2015, Shang et al., 2018]. Nowadays, more attention has been paid to neural-network-based approaches due to the successful application of deep learning in natural language generation. Most of the neural-network-based approaches [Rush et al., 2015, Chopra et al., 2016, Nallapati et al., 2016, See et al., 2017, Paulus et al., 2018, Wang et al., 2019] follow the sequence-to-sequence framework [Sutskever et al., 2014]. Especially, attention mechanisms [Bahdanau et al., 2015] are widely used in the sequence-to-sequence framework to improve the generation quality [Luong et al., 2015, Rush et al., 2015, Xu et al., 2015]. Self-supervised pretrained language models, such as BERT [Devlin et al., 2019], are also applied to natural language generation tasks including summarization. Liu and Lapata [2019] introduced a document-level encoder based on BERT which would better capture the semantics of a document. More recently, external knowledge in knowledge graphs and local semantic knowledge graphs are also used to improve the correctness of factual statements and quality in abstractive summarization [Zhu et al., 2020, Huang et al., 2020]. Also, multi-view seq2seq models are designed [Chen and Yang, 2020] by combining conversational structures from different views for better representation of human conversations.

### 6.2 Pragmatics

Pragmatics [Grice, 1975] has been studied in both linguistics and natural language processing for a long time. It generally studies the ways in which the context contributes to the meaning. Early approaches only consider cases or rule-based methods to evaluate pragmatics in language understanding and generation problems such as machine translation or dialogue systems [Rothkegel, 1986, Carberry, 1989, Iida et al., 1990]. Recent research focuses on using computational methods and automatic evaluation metrics in language games to evaluate the ability to infer through context [Frank and Goodman, 2012], which is usually called Rational Speech Acts (RSA) model. Wang et al. [2016] developed another language game and found that pragmatics models may not help the people who use less precise and consistent languages, as the pragmatics model assumes that the human is cooperative and behaving rationally. Fried et al. [2018] showed that explicit inference can also help learning-based RSA models.

Besides typical RSA models that committed to Grice's original target, there are also several other interesting directions to explore in computational ways. Kazemzadeh et al. [2014] introduced a way to evaluate pragmatics, where an image is used to identify an object inside. Then one player is asked to provide referring expression for the object while the other is asked to localize the object based on the expression. Many models have been developed for this task [Mao et al., 2016, Andreas and Klein, 2016, Monroe et al., 2017, Vedantam

et al., 2017]. In addition, Lewis et al. [2017] introduced a negotiation generation task that implicitly needs pragmatics inference in the learning system. Another data (or task) is called CommitmentBank, which evaluates the inference of a speaker's commitment towards the content of the complement under different entailment-canceling environments [Jiang and de Marneffe, 2019a,b]. Recently, Shen et al. [2019] improves text generation with techniques of computational pragmatics, which are comprised of information preservation and explicit modeling of distractors.

## 7. Conclusion

In this paper, we address the problem of out-of-context inference in dialogue summarization for the first time, and proposed Trans-KnowAttn to improve the inference ability of dialogue summarization. Based on our observation, such kind of inference is required for most of the dialogues in the DialSum dataset. Experiments demonstrate that the proposed model can outperform the previous state-of-the-art in terms of both traditional lexical metrics, and our newly proposed $F1_{CODC}$ significantly. We argue that with the help of out-of-context inference, neural models can better understand the pragmatics in human dialogues and thus improve the overall quality of summarization.

## Acknowledgment

The authors of this paper were supported by the NSFC Fund (U20B2053) from the NSFC of China, the RIF (R6020-19 and R6021-20) and the GRF (16211520) from RGC of Hong Kong, the MHKJFS (MHP/001/19) from ITC of Hong Kong, and the Tencent AI Lab Rhino-Bird Focused Research Program.

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

## Appendix A. Details of the Summarization Model

We present the details of the Knowledge Attention part of the knowledge-aware summarization model in this section.

**Encoder Formulation.** The encoder part is the same as what has been commonly used in previous neural summarization models. The input to the encoder is a sequence of dialogue word tokens $D = [w_1, w_2, \cdots, w_n]$, and the encoder produces a sequence of hidden states $\{\boldsymbol{h_1}, \cdots, , \boldsymbol{h_n}\}$.

**Decoder Formulation and Knowledge Attention.** In the decoding process, we include two attention channels, a standard attention that attends to encoder hidden states [Luong et al., 2015, Bahdanau et al., 2015] and a knowledge attention that attends to the retrieved knowledge list. Note that we infer knowledge in the word level and top $m$ candidates scored by the classifier will be retrieved. Thus we denote the list of knowledge word embeddings as $K = \{\boldsymbol{k}_1, \boldsymbol{k}_2, \cdots, \boldsymbol{k}_m\}$. For the $t$-th step of the decoder hiddent state $\boldsymbol{s}_t$, we use $\boldsymbol{s}_t$ to query the encoder hidden states as well as the list of knowledge vectors. Here we use $\beta_h(\boldsymbol{h}_i, \boldsymbol{s}_t)$ and $\beta_k(\boldsymbol{k}_i, \boldsymbol{s}_t)$ as scoring functions in the calculation of attention distribution for encoder hidden states and knowledge attention respectively, where $\beta$ can be an MLP or *general* function as in [Bahdanau et al., 2015]. The two attention based context vectors $\boldsymbol{h}_t^*$ and $\boldsymbol{k}_t^*$ are calculated as:

$$\boldsymbol{h}_t^* = \sum_i a_i^t \boldsymbol{h}_i \ \text{ and } \ \boldsymbol{k}_t^* = \sum_i v_i^t \boldsymbol{k}_i, \tag{5}$$

where the corresponding distributions are:

$$a_i^t = \text{softmax}_t(\exp(\beta_h(\boldsymbol{h}_i, \boldsymbol{s}_t))), \tag{6}$$

$$v_i^t = \text{softmax}_t(\exp(\beta_k(\boldsymbol{k}_i, \boldsymbol{s}_t))). \tag{7}$$

The knowledge context vector can be regarded as a fixed-size representation of the knowledge that has been inferred from the input dialogue. The context vectors $\boldsymbol{h}_t^*$ and $\boldsymbol{k}_t^*$ are concatenated with the decoder state $\boldsymbol{s}_t$ to produce the probability distribution over the vocabulary:

$$P_V = \text{softmax}(\boldsymbol{V}'(\boldsymbol{V}[\boldsymbol{s}_t, \boldsymbol{h}_t^*, \boldsymbol{k}_t^*] + \boldsymbol{b}) + \boldsymbol{b}'), \tag{8}$$

where $\boldsymbol{V}$, $\boldsymbol{V}'$, $\boldsymbol{b}$, and $\boldsymbol{b}'$ are learnable parameters. We follow See et al. [2017] to use two linear layers in this formulation.

The generation probability of copying from the input text $p_g \in [0, 1]$ is calculated as See et al. [2017]. To better facilitate the effects of the knowledge, we also apply a generation probability of knowledge $p_k \in [0, 1]$ as an indicator of copying concepts from the knowledge list. $p_g$ and $p_k$ for step $t$ is calculated from the encoder context vector $\boldsymbol{h}_t^*$, the knowledge context vector $\boldsymbol{k}_t^*$, the decoder state $\boldsymbol{s}_t$, and the decoder input $\boldsymbol{x}_t$:

$$p_g = \sigma(\boldsymbol{W}_{h*}^T \boldsymbol{h}_t^* + \boldsymbol{W}_s^T \boldsymbol{s}_t + \boldsymbol{W}_x^T \boldsymbol{x}_t + \boldsymbol{b}_g), \tag{9}$$

$$\begin{aligned} p_k = \sigma(&\boldsymbol{W}_{kh*}^T \boldsymbol{h}_t^* + \boldsymbol{W}_{ks}^T \boldsymbol{s}_t \\ &+ \boldsymbol{W}_{kx}^T \boldsymbol{x}_t + \boldsymbol{W}_k^T \boldsymbol{k}_t^* + \boldsymbol{b}_k), \end{aligned} \tag{10}$$

where matices $\boldsymbol{W}_{h^*}$, $\boldsymbol{W}_s$, $\boldsymbol{W}_x$, and $\boldsymbol{b}_g$ are learnable parameters for $p_g$, and $\boldsymbol{W}_{kh^*}$, $\boldsymbol{W}_{ks}$, $\boldsymbol{W}_{kx}$, $\boldsymbol{W}_k$, and $\boldsymbol{b}_k$ are learnable parameters for $p_k$. Here $\sigma$ is the sigmoid function.

Finally, both $p_g$ and $p_k$ are used to determine the probability of the next word to be generated. They serve as soft choices among sampling from $P_V$, copying a word from the input dialogue, or copying a word from the knowledge list, by sampling from the corresponding attention distribution:

$$
\begin{aligned}
P(w) = &\frac{1}{2}((1 - p_g)P_V(w) + p_g \sum_{i:w_i=w} a_i^t) \\
&+ \frac{1}{2}((1 - p_k)P_V(w) + p_k \sum_{i:w_i=w} v_i^t).
\end{aligned}
\tag{11}
$$

The loss function for time step $t$ is the negative log likelihood of the target word $w_t^*$:

$$
\mathrm{loss}_t = -\log P(w_t^*),
\tag{12}
$$

and the total loss for the input sequence is:

$$
\mathrm{loss} = \frac{1}{T} \sum_{t=1}^{T} \mathrm{loss}_t,
\tag{13}
$$

where $T$ is the total number of generated words.

## Appendix B. Case Studies

More case studies are presented in Figure 5.

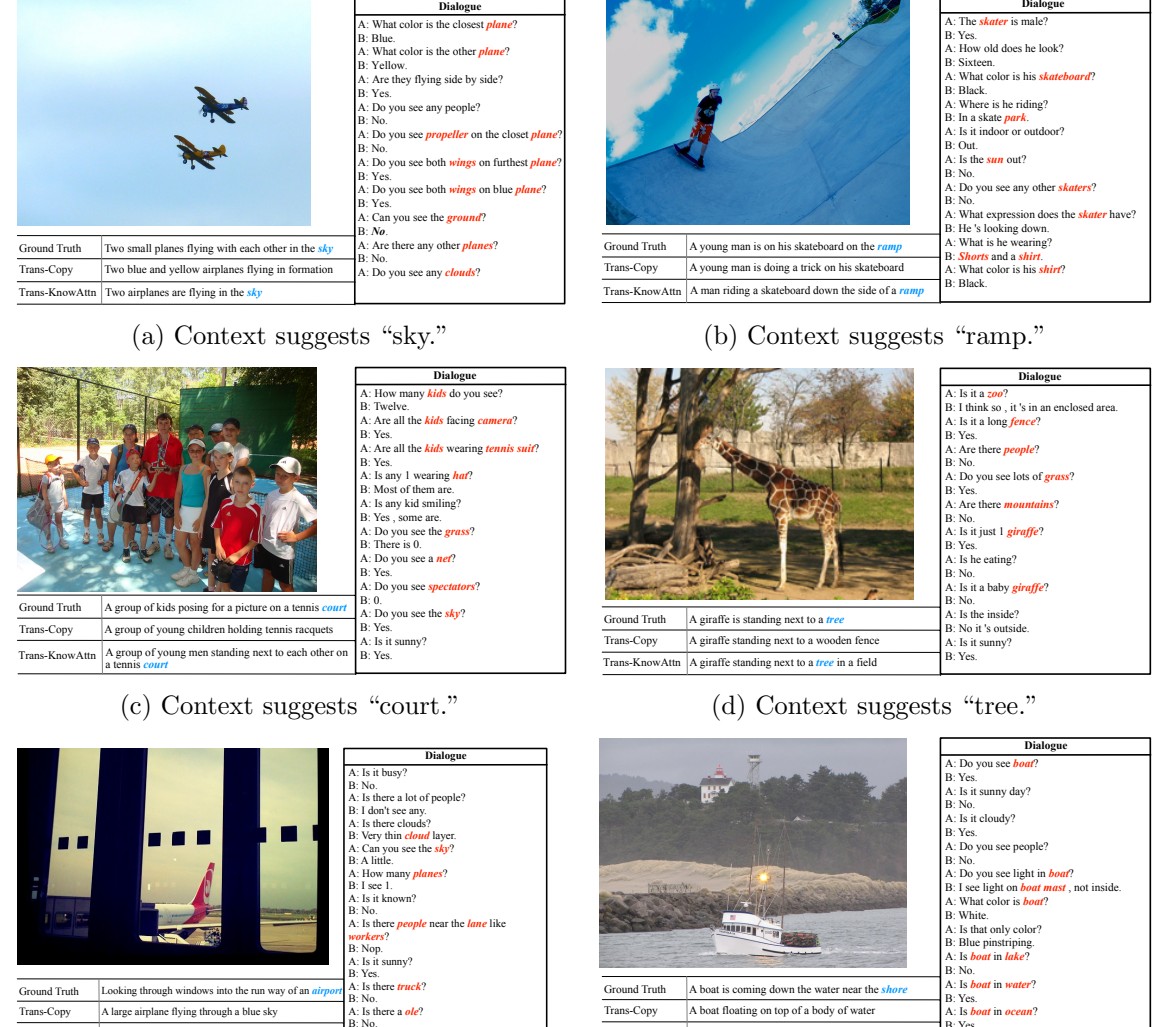

(a) Context suggests "sky."  (b) Context suggests "ramp."

(c) Context suggests "court."  (d) Context suggests "tree."

(e) Context suggests "airport."  (f) Context suggests "shore."

Figure 5: More cases of out-of-context inference. We highlight contextual concepts and conceptual inference with ***bold italics*** and ***bold italics*** respectively. Each figure is a comparison between Trans-Copy and +KnowAtten. For example, in (a), we will infer the picture of two airplanes flying in the sky by "*plane*" and "*No ground is seen.*" Trans-KnowAttn successfully inferred the concept "*sky*" while Trans-Copy didn't. In (c), from the "*tennis suit*," "*net*," and that the kids are *facing the camera*, we can infer that they are on a tennis court instead of other places. Also, Trans-KnowAttn successfully inferred the place where the kids are while Trans-Copy didn't.

