# OpenReview forum: "Do Boat and Ocean Suggest Beach? Dialogue Summarization with External Knowledge"
_AKBC.ws/2021/Conference — AKBC 2021_

### Official Review · Reviewer_sXci · 2021-07-16
**Well-written paper that identifies a new problem; model innovation is limited but overall a good paper**

**Rating:** 7
**Confidence:** 4

**Review:**

The paper tackles the problem of dialogue summarization, where a generated summary should include concepts that are not literally mentioned in the dialogue but can be inferred based on one’s commonsense. The paper provides precise definition of such concepts (called CODC in the paper) and evaluation metrics. It then proposes a new model that retrieves relevant concepts from ConceptNet, and feeds them to the text generator. Experiments show small improvements over a range of baselines in automatic evaluation, CODC evaluation, and manual human evaluation.

Strength
* The paper is generally well written and is easy to follow.
* The problem is generally well-motivated, and the definition of CODC and evaluation metrics are all reasonable.
* The model is clearly described and is well-executed.
* Experiments are comprehensive and useful, including a comparison with a range of baselines, ablations that show impact of different model components, and human evaluation.
* The paper includes good conclusions/discussions, e.g.,

Weakness:
* While I think the problem of CODC is generally well-motivated in the paper, it would still be good to see more evidence on whether inferencing CODCs is practically useful, i.e., whether humans actually prefer the summary with more CODCs. Human evaluation in the later part of the paper is not the fair evidence because crowdworkers were explicitly asked to consider the CODC aspect.
* Innovation in the model is limited. Specifically:
    * The specific way Knowledge Attention Network retrieves concepts is tightly coupled with the specific way CODC metrics are defined, which possibly overly favors Knowledge Attention Network in CODC metrics. For example, CODC metrics only consider new concepts which (1) appear in ConceptNet, and (2) are connected with concepts in the dialogue with depth of at most 4, which are fairly specific definitions. These definitions are exploited by the model based on Section 4.1 (e.g., in step 1, initial pool of concepts are retrieved based on the definition of CODC in Section 3).
    * Improvements over the baseline (Trans-Copy) are not substantial. Even based on a CODC metric (F1), the improvement is 0.6% absolute. (As a side note, I believe that the most important metric has to be F1, rather than precision or recall. High precision can be achieved by generating concepts as little as possible, and high recall can be achieved by generating concepts as many as possible.) Although the paper includes human evaluation, it is hard to solely rely on human evaluation results due to lack of details such as instructions and the agreement rate by crowdworkers.

Questions:
* During human evaluation, are annotators given groundtruth CODCs? Or are they only given the groundtruth summary and the generated summary?
* Based on the ablation, it looks like retrieval of concepts heavily relies on “co-occurrence”, which is a fairly naive method. I am curious to see the performance of the baseline which only uses co-occurrence, without leveraging the graph structure of ConceptNet. One specific way of implementing this baseline would be to keep a set of concepts from ConceptNet, and get the top-K concepts with the most co-occurrence with concepts in the dialogue. If this baseline is as good as the Knowledge Attention Network, it may mean that the role of “knowledge” in the model is limited.
* It would be useful to plot a graph of metrics (both standard metrics and F1_CODC) with varying numbers of CODCs at inference time. It would be ideal to see that improvements come from examples with many CODCs.

---

> ### Author Response · Authors · 2021-07-30
> **Response to review 3**
>
> Thanks for the review!
>
>
>
> - Human evaluation
>
>
>
> The annotators are not given ground-truth CODCs. They are only given the dialogue, ground-truth, and generated summaries. The annotators are not given suggestive questions and are asked to compare the two results fairly.
>
>
>
> - Ablation on Co-occurrence
>
> If we use Co-occurrence feature only for inferencing, the retrieved knowledge list can only reach F1_CODC score 11.23, compared to the figures in Table 2 (b), where when using all features the F1_CODC can reach 14.85. By feeding the retrieved knowledge using co-occurrence only to the summarization model, the resulting CIDEr is 102.89, compared with 104.00 when using all features. With these results, we can conclude that the CODC inference is not a naïve problem that can be solved using Co-occurrence solely and requires more sophisticated features like external knowledge from WordNet.
>
>
>
> - Figure with performance v.s. number of CODCs.
>
>
>
> The result is already shown in Figure 4 (a). The x-axis is the number of CODCs leveraged by the summarization model. The trend of the figures shows that within a certain threshold, the more inferred CODC the better the summarization performance.

---

### Official Review · Reviewer_EVVb · 2021-07-21
**Interesting task and a solid model and evaluation, but unclear if the dataset is a good fit for the task**

**Rating:** 6
**Confidence:** 3

**Review:**

This paper introduces a new dialogue summarization task that requires summaries to mention new concepts that are not explicitly mentioned in -- but may be implied by -- the dialogue. The task, CODC, repurposes the VisDial dataset (itself based on the MSCOCO captioning dataset), which involved information-seeking question-and-answer dialogues about an image, where the dialogue participants were prompted by the image's caption. The captions frequently mention some concepts that are not mentioned in, but are implied by, the dialogue. In CoDC, the task is to produce the reference caption (which is treated as a dialogue summary) from the dialogue (without using the images). Summarization models are evaluated on standard text overlap with the references, as well as an WordNet-based F1 metric that measures whether the model's output has inferred concepts in the references which were not mentioned in the dialogue.

The paper proposes a Trans-KnowAttn model, which uses the co-occurrences and word relatedness between concepts between dialogues and summaries in the training data to propose candidate concepts for a given dialogue, then attends to these concepts in a neural sequence-to-sequence abstractive summarization system. The proposed model outperforms a variety of reasonable neural sequence-to-sequence summarization baselines, as measured by both of the metrics above, and also in pairwise human judgements between Trans-KnowAttn and the best baseline system.

*Strengths*
1. Predicting inferrable knowledge (and more generally pragmatic implicatures, which this paper indirectly touches on) is an exciting and underexplored direction.
2. The model was well-motivated, combining a variety of sources of knowledge (co-occurrences in training data and lexical similarity) and approaches (classifiers and attentive neural models) in a reasonable and appropriate way. Ablations show that most of the components are valuable.
3. For a task that seems potentially very challenging (see weaknesses below for some questions about how difficult / well-defined the task is, as constructed using the dataset), the results seemed strong, with the proposed model outperforming a range of reasonable neural baselines.

*Weaknesses*
1. While I appreciated the motivation for the task, and that reusing VisDial allows this work to draw on a large body of work analyzing VisDial, I'm not totally convinced that VisDial is a good fit for this task. In the VisDial dataset collection, both dialogue participants saw the caption, and then produced the dialogue (with one participant seeing the image and answering questions asked by the other participant). The caption seems likely to have complementary information to the dialogue, but it's not totally clear to me that captions (or their novel concepts) are inferrable from the dialogue or can be reasonably called summaries. The example dialogues the paper shows did help a bit to convince me, but it's unclear whether these dialogues are cherry-picked. The paper would be much stronger if it evaluated the dataset quality with a human ceiling, by e.g. having people annotate summaries for some fraction of the dialogue and compute the CoDC metrics between these human annotations and the reference captions, or compute human inter-annotator CoDC.
2. While the paper was overall very clearly structured and easy to understand, a few details about the evaluation and models were unclear to me (see below), and there were a lot of minor writing issues throughout which could be fixed by a thorough proofreading (see below).

*Questions*
1. In Sec 3.1 CoDC definition, "shortest path between any pair of synsets": does distance use hypernym/hyponym relations, or synonym relations?
2. should $R_{CODC}$ also use max (as $P_{CODC}$ does)?
3. What is the copy mechanism used in Trans-Copy?
4. The performance of BertSum is surprisingly low. Was it fine-tuned on this dataset?
5. Are the hyperparameters of Trans-KnowAttn the same as Trans-Copy?
6. Human evaluation: how many times did annotators chose the neutral comparison option? Since these neutral results seem to be excluded from Table 2, it's unclear how many times Trans-KnowAttn was rated better than Trans-Copy, so it's especially hard to evaluate significance. It would help to do a statistical significance test here.
7. Can more details be given for the Randomization Test -- is this a permutation test? if so how was it performed?

*Suggestions*
- Be more specific about the CODC-based metric in the intro: does it compare against references?
- Figure 1 would be clearer with the role of images (they're just used in the VisDial construction) specified.
- Using new concepts is not exactly lexical entrainment, which (as defined by Brennan 1996) means two people coming to use the same terms when repeatedly discussing an object.
- "gold summarization $y$ is provided by an annotator" => it's a gold caption, not really a gold summary, and it was provided by an annotator in the captioning task.
- It would help to give more detail on how VisDial was constructed: both participants saw the caption, but only one person ("the answerer") saw the image, while the questioner is supposed to ask questions to find out more detail about the image (which often extends what's given in the caption).
- It's a bit difficult to reason about likely values for P_{CODC} and R_{CODC}, given the relatively complex (but reasonable) definitions of the functions, which rely on the structure of WordNet. Some example calculation of these or their reasonable ranges (e.g. give the statistics for an example dialogue) might be helpful to build intuition.
- A few more details should be given about the training procedure for clarity: I'm assuming the models need to generate the full reference caption (rather than just the CODC words), are trained to maximize log likelihood of captions given dialogues, and that "All five references are used in the training process" means a given dialogue is used in five different training instances, one time with each caption. This should all be clarified given more space though (especially if I'm misunderstanding).
- Unclear why using both GloVe and Word2Vec hurts performance, but could be due to some details about how the embeddings are combined (details on this are missing from the paper).
- Sec 6: "pragmatics may be hurt when people are using less precise and consistent languages". I'm unclear on how to interpret this: pragmatic modeling hurts the system? Pragmatic interpretations are difficult (for people? or systems?) when language is less precise and consistent?

*Typos*
- last para of page 2: extandable => extendable, "co-occurance" => "co-occurrence"
- "ability of inferencing" => "ability to infer"
- Sec 3: "by concept, we mean a noun word": noun word => noun phrase
- Sec 3: "most of examples" => "most examples"
- Sec 4: "while are out" => "while being out"
- "groud-truth" => "ground-truth"
- "typical similarity measurement" => "typical similarity measurements"
- "attend to knowledge embedding list" => "attends to the knowledge embedding list"
- "*a* copy mechanism is used"
- "from which human will easily inference" => "from which a human will easily infer"
- "fails to inference such new" => "fails to infer this new"
- "yield such inference" => "yield such an inference"

*** Update after response ***

Thanks to the authors for the response! Although I still think that the paper would be much stronger if it added a human ceiling (see suggestions in weakness 1 in my original review) to see how well-defined / challenging the dataset is, the response helped with my concerns, and I'm more positive about the paper. I've raised my score to a 6 (from a 5).

I would feel better about the framing if the captions weren't referred to as "summaries", since they're not summaries in the sense that's often used in NLP (written while referring to the document being summarized) and this may be a bit misleading to people (it was to me), since this really is a new task in a sense. But I think this could easily be changed.

Thanks too for the answers which clarified my other questions! It would also help to add a significance test to the human evaluation results.

---

> ### Author Response · Authors · 2021-07-30
> **Response to review 2**
>
> Thanks for the careful review.
>
>
>
> Response to Weakness:
>
>
>
> It is the fact that both participants are aware of the caption that contributes to our assumption that contributes to an inference gap between dialogues and captions. As captions are explicitly accessible to both participants, participants don't have to express the caption context explicitly in their conversations, and we hope to address this gap. The statistics on the dialogues where inference is needed are shown in Figure 2 (a) of our paper.
>
>
>
> Response to the questions:
>
> The paths use synonym relations to filter out synonyms.
>
> The R_CODC uses max already, while in a different way. We include the ground-truth summary where the generated summary can achieve the maximum recall in the numerator, as indicated in equation (3).
>
> The copy mechanism is the same as in Pointer Generator Network (PGN) [1], which has been introduced in the appendix.
>
> Yes, it’s fine-tuned on this dataset. BERT may not be good enough at such generation task where out-of-context inference is required.
>
> Yes, they are the same.
>
> For the 500 comparisons, 74 votes are neutral. votes favoring Trans-Copy, Trans-KnowAttn, and neutral: (206, 220, 74)
>
> Yes, it’s a permutation test. Denote the summaries generated by system A as {x_A_1, x_A_2, …, x_A_n} and system B as {x_B_1, x_B_2, …, x_B_n}. Denote the score generated by the specific evaluation metric, say, CIDEr, of system A as f(A). We would like to know whether f(A)-f(B) is significant. For each data entry with index i, permutate {x_A_i, x_B_i} to form different simulated systems A’ and B’, which yield 2^n different system pairs (A’, B’). Sample K permutations from all (A’ B’) pairs and calculate f(A’)-f(B’) for each of them. The f(A’)-f(B’) scores form a distribution histogram. The p-value can be simulated by checking the CDF position of f(A)-f(B) on the histogram.
>
>
>
> [1] Abigail See, Peter J. Liu, and Christopher D. Manning. Get to the point: Summarization
>
> with pointer-generator networks.

---

### Official Review · Reviewer_z3eg · 2021-07-22
**Can automated dialog summarization summarize concepts that are not immediately available in the input?**

**Rating:** 6
**Confidence:** 1

**Review:**

# Summary
The authors introduce a new problem, inferencing Concepts Out of DIalogue Context (CODC) wherein a ML model is tasked with producing concepts that are not explicitly mentioned in the input. The authors define this problem concretely and introduce metrics to study it. Particularly they define concepts as noun phrases that follow a very specific set of rules within wordnet. The authors propose metrics and a model to test the ability of systems on these CODC problems.

# Strengths
- The paper is well written.
- I like the rules that are clearly defined to be concepts. They are intuitive and make the phrase concepts seem less abstract.
- Adding human evaluations strengthens the paper.
- Modelling approaches are intuitive and well presented.

# Weakness
- This appears to be a very hard problem. The image is first "summarized" by dialog which must then be further summarized as a part of this setup. Since summarization almost guarantees a loss of information, I wonder how much scope there is to actually improve on the results on this dataset.
- The related work section does not do a good job of placing the work in the context of related works. The current work should be situated more clearly within the field as much as possible.
- I do not really like the phrase "concept" as it comes across as a very abstract term to me. Changing this is not critical but I think it would help to introduce the definition of what the author's mean exactly by 'concepts' asap in the text.
- The phrases "abstractive" and "extractive" models should be defined early on. I personally have never come across these phrases before. A simple sentence each should be sufficient.
- One line descriptions of the baseline  models in the main text would strengthen the paper.

# Other thoughts
- Figure 1 in the appendix helps develop the intuitions for the reader. Consider moving more examples from the paper in to the figure as it should enhance readability.

---

> ### Author Response · Authors · 2021-07-30
> **Response to review 1**
>
> Thanks for the review.
>
> - Related works
>
> The problem of inferencing out-of-context concepts is a new problem where no identical related works have been proposed. In this sense, we divide the related works section in our paper into two parts, dialogue summarization and pragmatics, where the first part is the concrete task in our work and the second one is where out-of-context inference is originated from. We believe this can provide enough background knowledge related to our work.
>
> - Concepts
>
> It’s been stated in the Introduction and the beginning of section 3.1 that we narrow down the definition of `concept’ to a noun word that satisfies specific restrictions using WordNet. It’s indeed valuable to have an overall definition at the beginning of the paper anyway. Thanks for pointing it out.
>
> - Abstractive and Extractive
>
> Abstractive and extractive summarization are two mainstream approaches for summarization models in NLP. We have already briefly explained the differences at the end of the first paragraph in the Introduction.
>
> - Baseline
>
> Sure, thanks for the suggestion. The one-line descriptions of baselines are in the appendix and we will move them to the main text if space permits.

---

### Decision · Program_Chairs · 2021-08-17

**Decision:**

Accept

**Comment:**

This paper introduces a new dialogue summarization task that requires summaries to mention new concepts that are not explicitly mentioned in -- but may be implied by -- the dialogue. The reviewers overall have a positive impression of the paper and agree that it presents an interesting challenge with compelling and complete modeling experiments. The camera-ready version of the paper would be strengthened by a human ceiling on the caption generation task. We also urge the authors to incorporate the reviewer's comments in the final version.